# Attention via Scattering Transforms for Segmentation of Small Intravascular Ultrasound Data Sets

**Lennart Bargsten**[1]                                                    LENNART.BARGSTEN@TUHH.DE

**Katharina A. Riedl**[2]                                                          K.RIEDL@UKE.DE

**Tobias Wissel**[3]                                                    TOBIAS.WISSEL@PHILIPS.COM

**Fabian J. Brunner**[2]                                                        FA.BRUNNER@UKE.DE

**Klaus Schaefers**[4]                                                 KLAUS.SCHAEFERS@PHILIPS.COM

**Michael Grass**[3]                                                    MICHAEL.GRASS@PHILIPS.COM

**Stefan Blankenberg**[2]                                      SEKRETARIAT-BLANKENBERG@UKE.DE

**Moritz Seiffert**[2]                                                            M.SEIFFERT@UKE.DE

**Alexander Schlaefer**[1]                                                      SCHLAEFER@TUHH.DE

[1] *Hamburg University of Technology, Institute of Medical Technology and Intelligent Systems, Hamburg, Germany*

[2] *Department of Cardiology, University Heart & Vascular Center Hamburg, Hamburg, Germany*

[3] *Philips Research - Hamburg, Germany*

[4] *Philips Research - Eindhoven, The Netherlands*

## Abstract

Using intracoronary imaging modalities like intravascular ultrasound (IVUS) has a positive impact on the results of percutaneous coronary interventions. Efficient extraction of important vessel metrics like lumen diameter, vessel wall thickness or plaque burden via automatic segmentation of IVUS images can improve the clinical workflow. State-of-the-art segmentation results are usually achieved by data-driven methods like convolutional neural networks (CNNs). However, clinical data sets are often rather small leading to extraction of image features which are not very meaningful and thus decreasing performance. This is also the case for some applications which inherently allow for only small amounts of available data, e.g., detection of diseases with extremely small prevalence or online-adaptation of an existing algorithm to individual patients.

In this work we investigate how integrating scattering transformations - as special forms of wavelet transformations - into CNNs could improve the extraction of meaningful features. To this end, we developed a novel network module which uses features of a scattering transform for an attention mechanism.

We observed that this approach improves the results of calcium segmentation up to $8.2\%$ (relatively) in terms of the Dice coefficient and $24.8\%$ in terms of the average Hausdorff distance. In the case of lumen and vessel wall segmentation, the improvements are up to $2.3\%$ (relatively) in terms of the Dice coefficient and $30.8\%$ in terms of the average Hausdorff distance.

Incorporating scattering transformations as a component of an attention block into CNNs improves the segmentation results on small IVUS segmentation data sets. In general, scattering transformations can help in situations where efficient feature extractors cannot be learned via the training data. This makes our attention module an interesting candidate for applications like few-shot learning for patient adaptation or detection of rare diseases.

**Keywords:** Deep learning, Intravascular ultrasound, Wavelets, Scattering transformation, Segmentation, Attention, Small data set, Calcifications, Lumen, Vessel wall

## 1. Introduction

Intracoronary imaging has proven to have a positive impact on percutaneous coronary interventions (Räber et al., 2018). A frequently used intracoronary imaging modality is intravascular ultrasound (IVUS). IVUS enables physicians to assess the morphology of vessel components like lumen, vessel wall and plaque distribution. Metrics like the lumen diameter, vessel wall thickness or plaque burden can be estimated by manually delineating the corresponding structures in multiple IVUS images. This is a considerably time-consuming process and the quality of the results depends strongly on the physician's experience. Automatic segmentation of IVUS images could therefore help to streamline the clinical workflow.

Automatic segmentation of IVUS images has already been studied in previous years. One can divide different segmentation approaches into two groups: non-data-driven methods and data-driven methods. Non-data-driven methods include active contour models, thresholding and gradient-based techniques (Katouzian et al., 2012; Balocco et al., 2014; Kermani and Ayatollahi, 2019). Data-driven methods require image annotations as additional source of information. State-of-the-art results are usually achieved by convolutional neural networks (CNNs) (Yang et al., 2019; Nandamuri et al., 2019; Xia et al., 2020) if the underlying training data sets are large enough. However, clinical data sets are often rather small. Furthermore, some applications inherently allow for only small amounts of available data, e.g., the detection of diseases with marginal prevalence or online-adaptation of an existing algorithm to individual patients. The convolutional filters, which are learned during training, are therefore often rather inefficient. This leads to extraction of features which are not meaningful and thus the performance decreases. In the medical domain it is hence crucial to develop methods which enhance CNN performance on small data sets. One way to do this is incorporating domain knowledge that takes into account certain properties of the data set and the task at hand (Karpatne et al., 2017; Bargsten and Schlaefer, 2020; Xie et al., 2021; Bargsten et al., 2021).

In this work, we want to investigate another approach: incorporating wavelet transformations into CNNs. Wavelets have previously been used for analysis of ultrasound and other medical images (Sudarshan et al., 2016; Prabusankarlal et al., 2016; Swarnalatha and Manikandan, 2020). An image is transformed into wavelet domain by convolving it with predefined filters. These predefined filters could be able to extract more meaningful features than a CNN is capable of when trained with only limited data. Wavelets have already been introduced to CNNs (Liu et al., 2019; Khatami et al., 2020; Sinha et al., 2020), mainly as a pre-processing step in order to feed the resulting features into the CNN. The resulting networks are referred to as *hybrid networks*. Another form of wavelet transformation is the scattering transformation (Mallat, 2012; Bruna and Mallat, 2013). Scattering transformations have been developed as cascaded feature extractors analogous to CNNs, but with defined filters. Some articles investigated scattering transforms as a pre-processing step to neural networks (Singh and Kingsbury, 2017a; Oyallon et al., 2017; Cotter and Kingsbury, 2019). So far, approaches involving scattering transformations have mainly been tested on natural image benchmark data sets like MNIST or CIFAR-10. An extensive analysis regarding the performance on medical image data is still pending.

In this work, we investigate whether incorporating scattering transformations into CNNs can increase their ability to extract meaningful features when trained with small IVUS

segmentation data sets. We tackle two different tasks: segmentation of calcifications as well as segmentation of lumen and vessel wall. In preliminary experiments, we observed that the already existing method of using scattering transformations as a pre-processing step to CNNs did not improve performance. We therefore propose a new method of fusing scattering transformations and neural networks. We integrate scattering transformations into a novel squeeze and excitation block (Hu et al., 2018; Roy et al., 2019), which we call *squeeze and excitation with scattering transform* (SEST). We test this approach with two different CNN segmentation architectures: U-Net (Ronneberger et al., 2015) and DeepLabV3 (Chen et al., 2017).

## 2. Material and Methods

### 2.1. Wavelet transformation

The wavelet transformation is frequently used in image analysis. It provides a change of data representation, analogous to the Fourier transformation. For $u \in \mathbb{R}^2$, a family of two-dimensional wavelet filters is defined as

$$\psi_{j,k}(u) = 2^{-j}\psi(2^{-j}R_{\theta_k}u) \tag{1}$$

with the mother wavelet $\psi(u)$, $j \in \{1, ..., J\} \subset \mathbb{N}$ and $R$ as a matrix defining a rotation of angle $\theta_k$ with $k \in \{1, ..., K\} \subset \mathbb{N}$. The individual wavelets are therefore obtained by scaling and rotating the mother wavelet. Analogous to Bruna and Mallat (2013) we denote $\lambda = (j, k)$ and $\Lambda_{J,K} = \{\lambda = (j, k) : j \in \{1, ..., J\}, k \in \{1, ..., K\}\}$. The wavelet transform of a 2D-signal $x(u)$ can be written as a set of convolutions

$$Wx(u) = \{x(u) * \phi_J, \ x(u) * \psi_\lambda(u)\}_{\lambda \in \Lambda_{J,K}} \tag{2}$$

with the scaling function $\phi_J = 2^{-J}\phi(2^{-J}u)$ serving as a low pass filter.

### 2.2. Scattering transformation

The scattering transformation can simplistically be described as a cascade of complex wavelet transformations with beneficial properties. These properties include translation invariance, stability to noise, stability to deformations and fast energy decay (Mallat, 2012). Energy decay means that the scattering coefficients decay to zero quite fast with increasing scattering order. An order of two usually proved to be sufficient (Bruna and Mallat, 2013).

To derive the corresponding equations, one defines an operator $U[\lambda]x = |x * \psi_\lambda|$ and a path $p = (\lambda_1, \lambda_2, ..., \lambda_m)$ representing a sequence of $\lambda$'s. $U$ can now be applied to $x$ with respect to the path $p$ via composition:

$$U[p]x = U[\lambda_m] ... U[\lambda_2] U[\lambda_1] x \tag{3}$$

$$= || ... |x * \psi_{\lambda_1}| * \psi_{\lambda_2}| ... * \psi_{\lambda_1}|. \tag{4}$$

The resulting descriptors are processed with the scaling function $\phi_J$ yielding the scattering coefficients:

$$S[p]x = U[p]x * \phi_J. \tag{5}$$

We now want to get the scattering coefficients of all possible paths. Let $\Lambda_{J,K}^m$ denote the set of all paths $p = (\lambda_1, \dots \lambda_m)$ of length $m$. Hence, $U[\Lambda_{J,K}^m]x = \{U[p]x\}_{p \in \Lambda_{J,K}^m}$ and $S[\Lambda_{J,K}^m]x = \{S[p]x\}_{p \in \Lambda_{J,K}^m}$. If one now defines the wavelet modulus propagator as (compare Equation 2)

$$\tilde{W}x(u) = \{x(u) * \phi_J(u), \ |x(u) * \psi_\lambda(u)|\}_{\lambda \in \Lambda_{J,K}}, \tag{6}$$

the scattering coefficients can finally be calculated via

$$\tilde{W}U[\Lambda_{J,K}^m]x = \{\tilde{W}U[p]x\}_{p \in \Lambda_{J,K}^m} \tag{7}$$

$$= \{S[\Lambda_{J,K}^m]x, \ U[\Lambda_{J,K}^{m+1}]x\}. \tag{8}$$

Originally, scattering transformations were introduced with the Morlet wavelet as underlying wavelet function. Performing scattering transformations with biorthogonal complex wavelets via a dual-tree complex wavelet transformation (DTCWT) (Kingsbury, 1999) was proposed in Singh and Kingsbury (2017b). In DTCWTs, real and imaginary part of the wavelets form a Hilbert pair and are processed in two separate trees with individual filters.

Cotter and Kingsbury (2019) fused DTCWT scattering transformations and CNNs by splitting the scattering orders into convolution-like layers - termed *invariant layers* - and adding mixing terms which can be learned during training. The authors provided an implementation for Pytorch (Cotter, 2019), which we also used for our studies. In their work, Cotter and Kingsbury (2019) inserted invariant layers before and after the first ordinary convolutional layer of a CNN but were not able to outperform other state-of-the-art CNNs on CIFAR-10. Nevertheless, they were able to reduce the number of parameters substantially.

## 2.3. Squeeze and excitation with scattering transformation

As mentioned previously, using DTCWT scattering transformations as a first or second layer of a CNN did not outperform state-of-the-art CNNs on CIFAR-10. Nevertheless, the scattering transformation is an efficient feature extractor and could therefore also be beneficial in other parts of a CNN. Based on the squeeze and excitation block (Hu et al., 2018; Roy et al., 2019) we developed an attention module for CNNs which makes use of the DTCWT scattering transform. Figure 1 depicts a corresponding sketch. The number of input feature maps $c_{in}$ is first reduced to one via a convolution with kernel size $1 \times 1$ yielding a condensed representation of the input. After that, the scattering transformation is applied. The result goes into two separate paths. One for spatial, the other for channel attention. The spatial attention path results in a single feature map with values between zero and one indicating unimportant and important regions of the input. The channel attention path results in $c_{in}$ neurons with values between zero and one indicating unimportant and important channels of the input. The results of both paths are multiplied with the input respectively. Finally, both results are combined by an elementwise max operation leading to the output with $c_{in}$ feature maps.

We call this block *squeeze and excitation with scattering transform* (SEST). It can be integrated into any CNN basically everywhere. In general, one can increase the number of output feature maps of the first convolution, e.g., to $c_{in}/k$ with $k \in \mathcal{N}$, but this can drastically increase the number of output feature maps of the scattering transformation

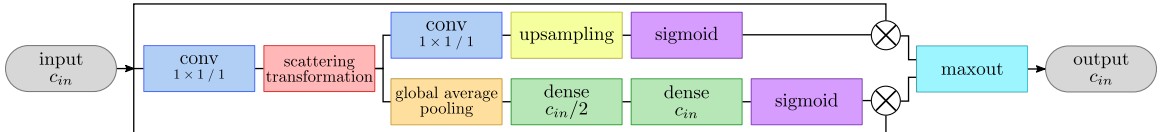

Figure 1: Sketch of the SEST block.

layer. For $c_{in}$ input channels, six different orientations of the underlying wavelet (which is the standard value for biorthogonal DTCWTs) and a scattering order of two, the number of output feature maps is $c_{out} = c_{in} \cdot 7^2 = c_{in} \cdot 49$ which can usually get very large in modern CNN architectures. We therefore restricted the number of output feature maps of the initial convolution of the SEST block to one. All scattering transforms in this work are of order two.

## 2.4. Data sets

We collected and investigated two different IVUS data sets. One for segmentation of calcifications, the other for lumen and vessel wall segmentation. We call these data set A and data set B respectively throughout the rest of this work. All images have a size of $500 \times 500$ pixels and were acquired with a 20 MHz phased array Eagle Eye Platinum probe (Philips Healthcare, San Diego, USA). Data set A consists of 432 images from 24 patients. Data set B comprises 410 images from 22 patients. Both data sets contain images with variable content like calcified an non-calcified plaques of various shapes, bifurcations, side branches, stents and guidewires. The annotations were made by an experienced cardiologist and are in the form of pixel masks. Figures 3 and 4 in the appendix show some images of both data sets with corresponding ground truth segmentation masks (first and second column).

## 2.5. CNN architectures

We integrated the methods explained above into two different state-of-the-art segmentation CNN architectures: U-Net (Ronneberger et al., 2015) and DeepLabV3 (Chen et al., 2017). We chose the U-Net to consist of residual blocks (He et al., 2016) analogous to (Milletari et al., 2016). For DeepLabV3 we used an adjusted ResNet (He et al., 2016) backbone. DeepLabV3 includes atrous (or dilated) convolutions (Yu and Koltun, 2016) as well as atrous spatial pyramid pooling (Chen et al., 2018). We tried to balance the capacity of both networks by adjusting the number of layers and feature maps such that they comprise roughly 12 M parameters. Both network architectures are depicted in Figure 2 (see appendix). In the corresponding baseline networks, scattering transformations were replaced with equivalent ordinary learnable convolutional layers. This means that the baseline CNNs have up to 8 k more parameters.

## 2.6. CNN training and evaluation

For every data set we investigated three different training set sizes while the test sets remained fixed in order to facilitate a reasonable comparison. The splits for data set A were

1. large: 318 training images and 114 test images,

2. mid: 145 training images and 114 test images,

3. small: 90 training images and 114 test images

and for data set B:

1. large: 289 training images and 121 test images,

2. mid: 146 training images and 121 test images,

3. small: 73 training images and 121 test images.

All images were resized to $256 \times 256$ pixels before feeding them into the CNNs. We performed online data-augmentation with random rotations and flips. We used the generalized Dice loss (Sudre et al., 2017) and employed the Adam optimizer with a learning rate of $\ell = 2e - 4$.

Networks of every method were trained 15 times with three-fold cross-validation for obtaining meaningful statistics. Every fold was trained for 200 epochs. After training, the model which performed best on the validation set was evaluated with the test set. As evaluation metrics we used the Dice coefficient for measuring the overlap as well as the average Hausdorff distance (Dubuisson and Jain, 1994) for measuring the edge alignment between ground truth and predicted segmentation masks. The average Hausdorff distance between two sets $A$ and $B$ is defined as

$$d_H^{ave} = \max \left( \underset{a \in A}{\mathrm{mean}} \, \underset{b \in B}{\min} \, d(a, b), \; \underset{b \in B}{\mathrm{mean}} \, \underset{a \in A}{\min} \, d(a, b) \right)$$

with the Euclidean distance $d(\cdot, \cdot)$. It introduces mean operations and is thus more suitable for comparing segmentation methods because it is less sensitive to outliers in contrast to the ordinary Hausdorff distance which uses max operations.

We performed t-tests for every comparison between SEST networks and the corresponding baseline CNNs. We term a result statistically significant, if $p < 0.01$.

## 3. Results and Discussion

Table 1 shows the segmentation results for calcifications. We can see that the less data is available, the better becomes the SEST approach in comparison to the baseline. Outperformances on the large data set are statistically significant but marginal. They increase for smaller data sets up to relative $8.2\%$ and $24.8\%$ for Dice coefficient and average Hausdorff distance respectively in the case of DeepLabV3. The performance of both networks are quite balanced. When trained with the smallest data set, DeepLabV3 seems to benefit more from incorporating SEST than U-Net Res. Additionally, the standard deviations of the baseline models trained with the small data set are substantially larger compared to the SEST networks. Hence, SEST seems to stabilize CNN training in this case.

The results on data set B are shown in Tables 2 and 3. Overall, it can clearly be seen that incorporating SEST turns out to be less beneficial than it was for data set A. Nevertheless, as with data set A the improvements via SEST tend to increase with decreasing data set size. In the case of vessel wall segmentation (2) U-Net Res with SEST is not able to outperform the

Table 1: Results for calcium segmentation.

| model | dset size | method | Dice coefficient [%] | | ave. Hausd. dist. [px] | |
|---|---|---|---|---|---|---|
| | | | value | p-value | value | p-value |
| U-Net Res | large | wavelet | $66.94 \pm 0.48$ | $< 0.01$ | $7.92 \pm 0.60$ | $< 0.01$ |
| | | baseline | $66.20 \pm 0.90$ | | $8.89 \pm 1.12$ | |
| | mid | wavelet | $64.74 \pm 0.69$ | $< 0.01$ | $10.32 \pm 1.06$ | $< 0.01$ |
| | | baseline | $63.69 \pm 0.96$ | | $11.96 \pm 1.29$ | |
| | small | wavelet | $58.62 \pm 1.46$ | $< 0.01$ | $13.31 \pm 1.20$ | $< 0.01$ |
| | | baseline | $55.38 \pm 2.99$ | | $16.89 \pm 3.45$ | |
| DeepLabV3 | large | wavelet | $66.57 \pm 0.34$ | $< 0.01$ | $8.09 \pm 0.48$ | $< 0.01$ |
| | | baseline | $66.00 \pm 0.42$ | | $8.61 \pm 0.50$ | |
| | mid | wavelet | $62.36 \pm 0.43$ | $< 0.01$ | $11.83 \pm 0.68$ | $< 0.01$ |
| | | baseline | $61.38 \pm 0.77$ | | $12.93 \pm 0.94$ | |
| | small | wavelet | $59.71 \pm 0.95$ | $< 0.01$ | $11.51 \pm 0.88$ | $< 0.01$ |
| | | baseline | $55.18 \pm 3.86$ | | $15.30 \pm 4.56$ | |

Table 2: Results for vessel wall segmentation.

| model | dset size | method | Dice coefficient [%] | | ave. Hausd. dist. [px] | |
|---|---|---|---|---|---|---|
| | | | value | p-value | value | p-value |
| U-Net Res | large | wavelet | $79.51 \pm 0.57$ | $0.14$ | $1.85 \pm 0.10$ | $< 0.01$ |
| | | baseline | $79.29 \pm 0.50$ | | $2.03 \pm 0.11$ | |
| | mid | wavelet | $76.15 \pm 0.61$ | $0.08$ | $2.56 \pm 0.18$ | $0.02$ |
| | | baseline | $75.86 \pm 0.47$ | | $2.69 \pm 0.15$ | |
| | small | wavelet | $67.05 \pm 1.23$ | $0.19$ | $4.80 \pm 0.44$ | $< 0.01$ |
| | | baseline | $66.63 \pm 1.39$ | | $5.36 \pm 0.43$ | |
| DeepLabV3 | large | wavelet | $77.86 \pm 0.52$ | $0.33$ | $2.16 \pm 0.14$ | $0.31$ |
| | | baseline | $77.78 \pm 0.49$ | | $2.18 \pm 0.08$ | |
| | mid | wavelet | $74.44 \pm 0.58$ | $< 0.01$ | $2.69 \pm 0.16$ | $0.03$ |
| | | baseline | $73.79 \pm 0.72$ | | $2.85 \pm 0.27$ | |
| | small | wavelet | $62.20 \pm 1.30$ | $< 0.01$ | $5.07 \pm 0.25$ | $< 0.01$ |
| | | baseline | $60.91 \pm 1.37$ | | $5.80 \pm 0.48$ | |

baseline significantly by means of the Dice coefficient. Only the Hausdorff distance shows statistically significant but minor improvements. Interestingly, DeepLabV3 does overall perform a little bit worse than U-Net Res. The same basically holds for lumen segmentation. For both networks, the improvements through SEST are statistically significant only when trained with the small data set. We can therefore infer that the benefits by SEST are greater for segmentation of small structures such as calcifications than for segmentation of larger structures like lumen and vessel wall. Figures 3 and 4 (appendix) show some examples of predicted segmentation masks.

## 4. Conclusion

In this work we presented a new approach of incorporating scattering transformations into convolutional neural networks (CNNs). We developed the *squeeze and excitation with scat-*

Table 3: Results for lumen segmentation.

| model | dset size | method | Dice coefficient [%] | | ave. Hausd. dist. [px] | |
|---|---|---|---|---|---|---|
| | | | value | p-value | value | p-value |
| U-Net Res | large | wavelet | $90.54 \pm 0.36$ | 0.39 | $1.40 \pm 0.17$ | 0.05 |
| | | baseline | $90.50 \pm 0.34$ | | $1.51 \pm 0.17$ | |
| | mid | wavelet | $90.12 \pm 0.60$ | 0.68 | $1.38 \pm 0.19$ | 0.09 |
| | | baseline | $90.22 \pm 0.49$ | | $1.53 \pm 0.39$ | |
| | small | wavelet | $86.59 \pm 1.07$ | **< 0.01** | $2.94 \pm 0.97$ | **< 0.01** |
| | | baseline | $84.63 \pm 0.58$ | | $4.25 \pm 0.56$ | |
| DeepLabV3 | large | wavelet | $90.12 \pm 0.41$ | 0.07 | $2.16 \pm 0.14$ | 0.07 |
| | | baseline | $89.93 \pm 0.26$ | | $1.57 \pm 0.13$ | |
| | mid | wavelet | $89.81 \pm 0.41$ | 0.01 | $1.40 \pm 0.20$ | 0.06 |
| | | baseline | $89.44 \pm 0.46$ | | $1.58 \pm 0.38$ | |
| | small | wavelet | $84.61 \pm 1.06$ | **< 0.01** | $3.05 \pm 0.83$ | **< 0.01** |
| | | baseline | $83.51 \pm 0.83$ | | $3.96 \pm 0.86$ | |

*tering transform* (SEST) module which can flexibly be inserted at various positions into any CNN. The performed studies on intravascular ultrasound (IVUS) image segmentation showed that CNNs comprising the SEST module outperform similar CNNs without the SEST module when trained on small data sets. This supports our assumption that the features extracted by the scattering transformation are quite meaningful for CNNs when dealing with data scarcity. Furthermore, it turned out that the benefits of SEST for IVUS segmentation are particularly high when targeting small structures like calcifications.

Future research could focus on integrating scattering transformations into CNNs for processing image data of other modalities like computed tomography, magnetic resonance imaging, optical coherence tomography or magnetic particle imaging. Additionally, the shown improved performance on small data sets could make SEST or other methods of scattering transformation and CNN fusion promising candidates for few-shot learning tasks like patient adaptation or detection of diseases with small prevalence.

## Acknowledgments

This work was partially funded by the European Regional Development Fund (ERDF) and the Free and Hanseatic City of Hamburg in the Hamburgische Investitions- und Förderbank (IFB)-Program PROFI Transfer Plus under grant MALEKA.

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

## Appendix A. CNN SEST architectures

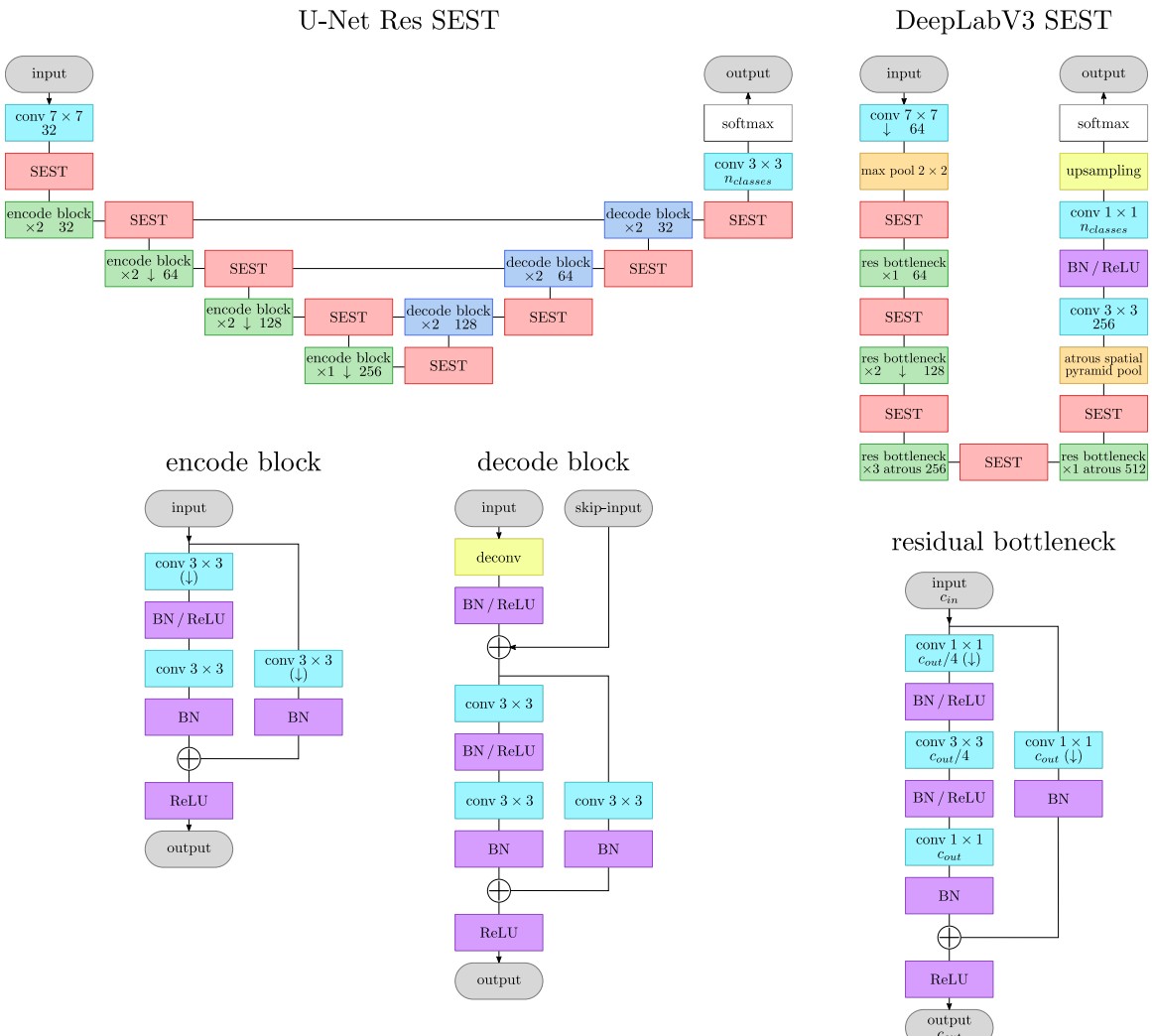

Figure 2: Sketches of U-Net Res and DeepLabV3 extended with SEST blocks.

## Appendix B. Exemplary segmentation images

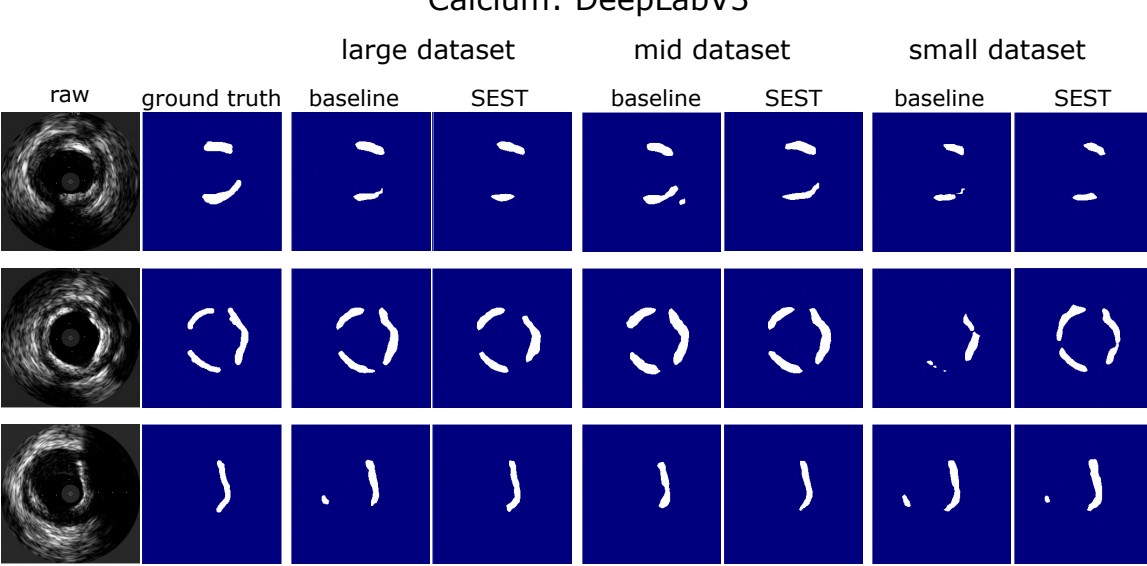

Figure 3: Exemplary calcium segmentations divided into results from U-Net Res and DeepLabV3.

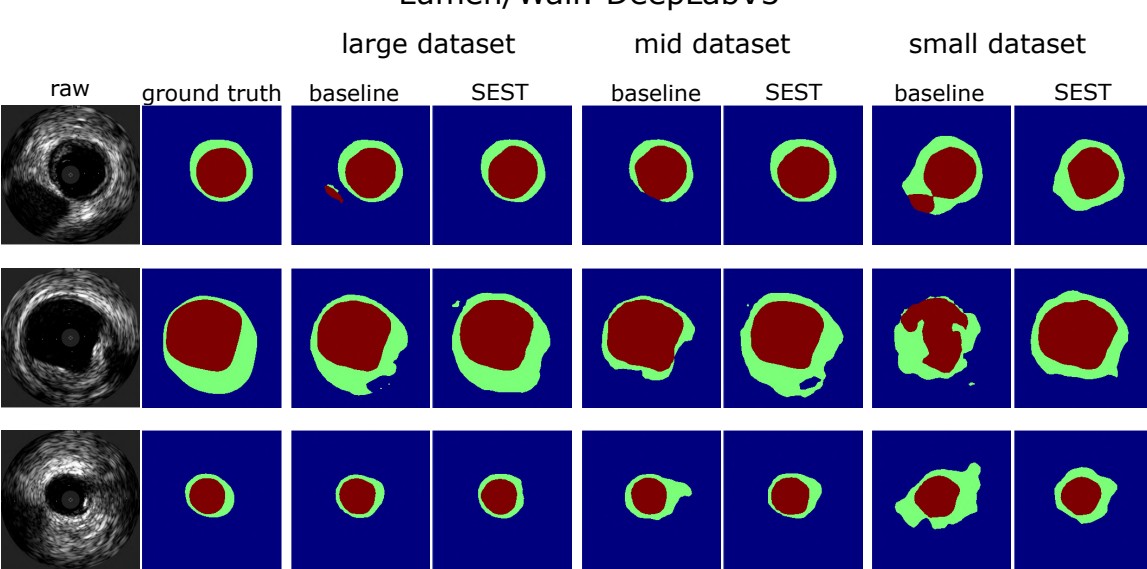

Figure 4: Exemplary lumen/vessel wall segmentations divided into results from U-Net Res and DeepLabV3.

