# OpenReview forum: "Attention via Scattering Transforms for Segmentation of Small Intravascular Ultrasound Data Sets"
_MIDL.io/2021/Conference — MIDL 2021_

### Official Review · AnonReviewer1 · 2021-03-08

**Confidence:** 5
**Preliminary Rating:** 3
**Recommendation:** Poster
**Final Rating:** 3

**Summary:**

This paper proposes the use of Scattering transforms in CNNs for the segmentation of small datasets. The existing “squeeze and excitation” block is modified to make use of wavelet decomposition (scattering transform) and is incorporated into standard segmentation networks (UNet and DeepLabV3). The developed models are applied to the tasks of segmentation of lumen and vessel walls in intravascular ultrasounds images.


**Strengths:**

The paper is well written and technically sound. A good description of the related work and of the method are provided. Promising results are particularly obtained on small datasets with a good discussion.


**Weaknesses:**

The improvements are good for small datasets. Very small improvements are obtained when using the full size of the dataset (the t-test is incorrect for the proposed evaluation, see detailed comments). The motivation of the approach for IVUS is therefore questionable.


**Deanonymize Review:**

no

**Detailed Comments:**

I suggest several modifications to improve the quality as stated below:

1- The abstract should be re-structured. The motivation, with mention of scattering transform being well suited to small datasets, should not be in the last paragraph. In the 1st paragraph: “leading to extraction of image features which are not…” It leads to poor segmentations and, in turn, not meaningful features.

2- The motivation is rather good except in 2.3: It didn’t work in early layers, but it could work anyway.

3- “online-adaptation of an existing algorithm to individual patients” seems a different problem. I wouldn’t mention it in the paper.

4- In introduction: “State-of-the-art results are usually achieved by CNNs”: specify that you mean on large datasets, otherwise the following sentences are contradicting.

5- In 2.3: I don’t get “one can increase the number of output feature maps of the first convolution, e.g., to c_in/k”.

6- In 2.3. Did you test with other orders (order two)? Why did you choose this order.

7- Can you specify the number of parameters of the baseline and the computation time of the three networks?

8- The t-test is incorrect in this scenario. You artificially increase the statistical power by adding more repetitions. The more train/val splits you use, the more statistically different your results will be.

9- You motivate the method for small datasets in IVUS, but you have sufficient data for obtaining good results with the baseline and you have to simulate the lack of data. You should comment on this, particularly since the statistical test is incorrect and the means are very close for the full size dataset.

10- 2.6 Specify what interpolation is used (both for images and GTs)

11- Why not using something like an early stopping on the validation loss?

12- In conclusion, you mention other modalities, comment on how it would scale to 3D data.


**Final Rating Justification:**

Thank you for your response. Some comments have been answered.
Regarding the t-test (my comments 8 and 9), I still think it is incorrect. From what I understand now, which I think is not clear in the paper, is that you use only one k-fold split and evaluate different (15) initializations of the methods. This way, or using 15 different splits, are both incorrect to show the significant difference. It gives the illusion of statistical significance. By increasing the number of repetitions in your case, anything can be made significantly different. You test the difference due to initialization, which is not relevant. What the test should evaluate is the difference due to variation in sampling of the population.
I strongly advise to revise this part.
I keep my previous rating.

**Justification Of The Preliminary Rating:**

The method is interesting and seems technically sound. Yet the results do not fully support the assumptions/motivation since artificial reduction of the dataset is needed to observe good improvements in comparison to the baseline. I still suggest a poster presentation if the comments can be addressed.


**Paper Type:**

methodological development

**Questions To Address In The Rebuttal:**

The comments above should be addressed.

**Special Issue:**

no

---

> ### Author Response · Authors · 2021-03-16
> **Answers to Comments of Reviewer #1**
>
> We thank Reviewer #1 for his detailed and concise feedback.
>
>
> 0) We agree that data availability for common tasks related to IVUS, such as assessment of stenoses and stent fit, is principally not very limited. However, particularly demanding and complex cases occur rather rarely, like severe plaque burden at bifurcations. Our method could help to increase the performance on such cases where only very few labeled images are available.
>
>
> 1) The statement that scattering transformations are well-suited for small datasets can only be made after the presentation of our results. Before that, it is just a hypothesis. We therefore feel that the abstract’s structure is okay. Moreover, it is more like that not meaningful features lead to poor segmentation results, not vice versa.
>
>
> 2) We are not sure if we understand the statement. Using scattering transforms as a preprocessing step for CNNs did not lead to improved performances on CIFAR-10 [Cotter and Kingsbury (2019)]. In the introduction, we also mentioned that this approach did not lead to performance improvements for our problem either.
>
>
> 3) Indeed, there are many approaches of dealing with patient adaptation. We think that a method which is able to extract meaningful features from only a small amount of individual patient data could help improving approaches which rely on efficient latent space encoding.
>
>
> 4) We agree and adapted the corresponding sentence.
>
>
> 5) This statement is most easily explained with Figure 1. It can be seen that the first convolutional layer of the SEST block has a single output feature map, thus aggregating the input into a single channel. This feature map is then scattering transformed. It is possible to feed multiple channels/feature maps into the scattering transformation. But this drastically increases the number of output feature maps of the scattering transformation.
>
>
> 6) We indeed tested orders of one, two and three. Whereas a single order led to unusable results, an order of three led to extremely large networks which did not perform better than using an order of two. We therefore chose an order of two for our experiments.
>
>
> 7) The U-Net and DeebLabV3 SEST models had 11.955 M  and 12.032 M parameters, respectively. The baseline models had up to 8 k parameters more, since the scattering transformation layers were replaced with learnable convolutional layers. A cross-validation run with the large dataset took around 75 minutes. Only 15 minutes for the smallest dataset. The training times for SEST networks and baseline CNNs were the same. All experiments were performed on an NVIDIA Titan RTX GPU.
>
>
> 8) We do not agree with this statement and try to explain the situation in more detail. For every model and dataset size, we performed three-fold cross-validation. We repeated this procedure 15 times. The training set split as well as the test set remained the same. The variations of results throughout the different repetitions therefore originated solely from different network initializations, not from the data. This means that the 15 samples are independent due to different random initializations of the network. We therefore believe that our approach is correct.
>
>
> 9) Again, we believe our statistical results to be correct. Furthermore, for studying and showing the limits of our method, we think that artificially decreasing the size of a larger dataset is a meaningful and controlled approach. As already stated in 0) and 3), we think that a method which is able to extract meaningful features from only a small amount of data could help improving performance on, e.g., detection of rate diseases or patient adaptation.
>
>
> 10) We used bilinear interpolation for all images and nearest neighbor interpolation for the segmentation masks.
>
>
> 11) Early stopping is indeed an appropriate method. However, it could lead to poor performance, if the network stucks on a plateau for multiple epochs before further reducing the loss. Using the model which performed best on the validation set for testing, as we did, is also appropriate. Nevertheless, it requires training the network for a sufficient amount of epochs which has to be found beforehand. Therefore, one usually considers a rather large margin of epochs which basically leads to longer training times than early stopping.
>
>
> 12) This is a good question. Generally, one can apply scattering transformations to volumetric image data, too. Due to the curse of dimensions this would lead to rather large networks. Hence, the image size would have to be reduced. It remains the question if the resulting scattering transformations would provide meaningful features for small 3D crops. This would be interesting to investigate in further studies.
>
>
> 13) Finally, we want to stress that we see an artificial reduction of dataset size for investigating a new method not as a problem but as a scientific and reasonable way of approaching such studies.
>
>
> We again thank Reviewer #1 for his fruitful comments.

---

> ### Author Response · Authors · 2021-03-23
> **Statistical Testing**
>
> We get the point you are making and we agree that training with multiple different CV splits can increase explanatory power of the results. However, our goal was to reduce the dependence of the results on different network initializations. The smaller the dataset, the larger the dependence on network initializations and thus the larger the variance in results. As an example: we train both the baseline CNN (A) and our new method (B) twice and obtain the following results (Dice only for simplicity): $A_1 = 60$ %, $A_2 = 65$ %, $B_1=66$ %,  $B_2 = 70$ %. These values indeed reflect some results we obtained in our study. Two points follow from these results:
>
> 1. The variance in results due to different network initializations is quite high and can't be ignored.
>
> 2. Wich results should be picked, if repeating the experiments  for averaging out the influence of different network intializations, as you say, is not correct? One could choose $A_1$ and $B_2$ and report large improvements or $A_2$ and $B_1$ and report marginal improvements. Neither approach makes much sense, in our opinion.
>
> If only multiple CV splits were trained without repetitions, the variance due to different network initializations would remain in the results and lead to biases.
>
> Finally, we think that an approach which varies CV splits (averaging out different data samples) and repeats training with every CV split multiple times (averaging out different network initializations) gives most meaningful results and statistics. However, this approach is extremely time-consuming and often not feasible when training neural networks.
>
> Thanks again for your remarks.

---

### Official Review · AnonReviewer3 · 2021-03-08

**Confidence:** 4
**Preliminary Rating:** 3
**Recommendation:** Oral
**Final Rating:** 3

**Summary:**

The team suggests a simple, elegant approach for image segmentation, particularly effective with medium size datasets.
The proposed approach integrates wavelet transforms in CNNs, via an attention mechanism which relies on the well known squeeze and excite module. The experiments show that the adopted CNNs for segmentation benefits more when less training data are used, which seems to be very promising moving towards few shot learning problems.

**Strengths:**

The paper is very well written and it is a pleasure to follow.
Using wavelet transform in combination with CNN is not the novelty of the paper as stated by the authors. Similarly, is not novel the use of attention mechanisms in segmentation networks (see Oktay 2018), it is however novel the way these feature are integrated in the proposed SEST block.The authors conduct multiple experiments to test the method - use multiple architecture and explore different training set size scenarios.

**Weaknesses:**

Part of the new block relies on a Pytorch implementation from Cotter et al 2019 - which credits the importance of open-sourcing code. The authors should also consider open sourcing their code to facilitate reproduction and application of their SEST block.

Although the experiments were designed to assess the added value of SEST in small dataset regimes, a dataset with 70 annotated images in some circumstances remain a luxury.

Online augmentation with random rotations and flips was done, but there are no details about this augmentation process, which - considering ultrasounds, it can be a reason for which some of the network used in the study might have underperformed.

A section with limitations and contributions is missing.

**Deanonymize Review:**

no

**Detailed Comments:**

The authors explain that they previously tested how CNN could benefit from Wavelet Trasform as a pre-processing operation, with no satisfactory results. It would be interesting to know more about the way these features were used.

A curiosity is whether it has been tried to concatenate these features to the input of the network and fuse these features within the network using either a convolutional kernel or an attention mechanism or if any other strategy was investigated.

The authors put emphasis on the fact DeepLabV3 underperformed U-Net Res - it would definitely be interesting to receive their opinion of what has caused this surprising - as they say - result.

Online data-augmentation with random rotations and flips might harm network training considering that ultrasound is a directional imaging modality, what range of rotation was used?

**Final Rating Justification:**

I think the claim for few shot learning is a bit excessive and can be misleading/unfair to works that really demonstrate their approaches excel with using few data. In my opinion, an extension of this work or the camera ready should not lack of an experimental section where the number of training samples is reduced to the minimum necessary, to empirically clarify if the term few-shot learning is appropriate here.

**Justification Of The Preliminary Rating:**

The idea is very interesting and improving performance of segmentation networks with really few data samples is of outmost importance at this stage. However the study does not really show how few samples can be processed to obtain good quality segmentations. There are medical imaging segmentation problems where the shape is very peculiar (e.g. shoulder, cartilage tissues) and it is hard to imagine precise and fine segmentation outputs with limited training data.

**Paper Type:**

both

**Questions To Address In The Rebuttal:**

The authors should show how much the training set can be reduced in size until the algorithm cease to provide meaningful improvements over the baseline approaches.
Since the attention mechanisms have potential to improve image segmentation regardless of the wavelet transform, additional comparative networks with attention mechanisms should be used to further assess the importance of the wavelet transform.

**Special Issue:**

yes

---

> ### Author Response · Authors · 2021-03-16
> **Answers to Comments of Reviewer #3**
>
> We thank Reviewer #3 for his comments and recommendations.
>
>
> 1) We agree that publishing code is great for making research fast and efficient. However, this work originated in the framework of a project and the rights regarding a publication of data and code have not yet been clarified. Open sourcing our code is therefore currently not possible.
>
>
> 2) We tested our approach on three different dataset sizes which we considered clinically relevant. Due to time and space constraints, we did not further decrease the dataset size. However, the presented results indicate a clear trend towards larger improvements of our method compared to the baseline.
>
>
> 3) We agree that data augmentation is very important to achieve good results. We therefore performed on-the-fly data augmentation with random rotations and flips. Due to the central view, there are no preferred directions in IVUS images if they are represented in Cartesian coordinates (compare images in appendix). The probe itself is circular symmetric. The range of rotation angles is therefore not limited and the angles were sampled from $[0, 2\pi)$. The same holds for flips, which can be horizontal and/or vertical. One could also consider transforming the images into polar representation, such that the horizontal direction indicates the ultrasound beam angle and the vertical direction indicates the penetration depth. In this case, only horizontal flips and horizontal (circular) translations would make sense.
>
>
> 4) Given the limited space, we opted against a separate contributions and limitations section. These are mentioned in the introduction as well as the results and discussion section, respectively. We list them here:
> Contributions: Our contribution is a novel CNN block (SEST) comprising a scattering transformation layer to enhance attention mechanisms. We tested this new block on intravascular ultrasound image data and showed that it increases calcium segmentation performance as well as vessel wall and lumen segmentation performance.
> Limitations: The improvements decrease for larger datasets. Smaller structures like calcifications benefit more from the SEST block than larger structures like lumen and vessel wall.
>
>
> 5) We investigated how the scattering transformation performed as a pre-processing step to the CNN. This means that the actual input image was scattering transformed, yielding $7^2 = 49$ or even $7^3 = 343$ feature maps. These feature maps were then fed into the CNN.
>
>
> 6) A concatenation of the transformed image with the actual input image does not seem very intuitive, since the scattering transform contains downscaled versions of the input image (through the scaling function). However, it would be interesting to develop and study further approaches of integrating wavelets or scattering transformations into CNNs. The authors of [1, 2], e.g., use multiple stages of a wavelet transformation as inputs of deeper network layers.
>
>
> 7) Usually, papers on segmentation report improved performance of DeepLabV3 compared to U-Net. However, we suppose that these cases do often times not deal with small datasets. So far, we always observed improved performance of DeepLabV3 if datasets tend to be large. It seems that U-Net is able to extract more meaningful features from less data than DeepLabV3. We think that the skip connections could play an important role in this case. Nevertheless, more experiments are required to verify these statements.
>
>
> 8) See 3)
>
>
> 9) See 2) regarding dataset sizes. Finally, we want to stress that our baseline networks indeed used an attention mechanism. But with ordinary learnable convolutional layers instead of scattering transformations. We mentioned that in section 2.5. This means that the comparison was fair. The baseline networks even had up to 8 k parameters more than the SEST networks. We clarified this by adjusting the following paragraph in section 2.5 of our manuscript:
> “In the corresponding baseline networks, scattering transformations were replaced with equivalent ordinary learnable convolutional layers. This means that the baseline CNNs have up to 8 k more parameters.”
>
>
> We again thank Reviewer #3 for his valuable recommendations.
>
>
>
> References:
>
> [1] Fujieda, S., Takayama, K., & Hachisuka, T. (2018). Wavelet convolutional neural networks. ArXiv 1805.08620. https://arxiv.org/abs/1805.08620
>
>
> [2] Liu, P., Zhang, H., Lian, W., & Zuo, W. (2019). Multi-Level Wavelet Convolutional Neural Networks. IEEE Access, 7, 74973–74985. https://doi.org/10.1109/ACCESS.2019.2921451

---

### Official Review · AnonReviewer4 · 2021-03-08

**Confidence:** 3
**Preliminary Rating:** 2
**Recommendation:** Poster
**Final Rating:** 3

**Summary:**

The paper presents a novel neural network building block that combines scattering transform and squeeze-and-excitation networks. The main motivation is that for some problems where not enough training data is available a lot of the network capacity is wasted to learn basic features. The authors propose to introduce prior knowledge by incorporating scattering transform into the convolutional blocks. The method is applied to intravascular ultrasound images segmentation.

**Strengths:**

- It is an interesting idea to combine scattering transform and squeeze-and-excitation networks.
- The paper is clearly written and easy to follow.
- It is good that the authors target problems where data driven approaches can be applied for small datasets.

**Weaknesses:**

Since the paper presents novel methodology, it would have been better to see more experiments. It would be a lot more persuading if the authors present results for variants of their method that exclude some of the introduced components (so-called ablation experiments).

While the authors made an effort to balance the number of parameters between the two different networks that use the newly introduced SEST blocks (both the UNet and the DeepLabV3 networks have around 12M parameters), it is unclear if they made the same effort of balancing the number of parameters between the baseline and the proposed architectures.

**Deanonymize Review:**

no

**Detailed Comments:**

Minor comment:
In the introduction the authors discuss shape techniques as non-data driven methods, but thy can arguably be placed in data-driven category as they rely on statistical shape models.

**Final Rating Justification:**

The authors have partially addressed my concerns, specifically about the number of parameters of the baseline that was of the most concern.

**Justification Of The Preliminary Rating:**

As mentioned above, I think the paper presents an interesting idea. However, given the presented experiments I cannot fully judge if the claimed improvements in performance are due to different network capacity or the novel methodology.

**Paper Type:**

methodological development

**Questions To Address In The Rebuttal:**

The authors should specify the number of parameters in the baseline. Without that it is not possible to see if the improvements of the performance, which are minor overall, are due to the novel architecture or due to larger network capacity.

Ablation studies will also go a long way towards making the case for the proposed architecture.



**Special Issue:**

no

---

> ### Author Response · Authors · 2021-03-16
> **Answers to Comments of Reviewer #4**
>
> We thank Reviewer #4 for his valuable remarks.
>
>
> 1) We agree that ablation studies can provide additional insight, and we have indeed tested different variations of the architecture. However, in this paper we focus on different dataset sizes as another parameter.
>
>
> 2) As mentioned in section 2.5, in the case of baseline CNNs, we replaced the scattering transformation layer inside the SEST block with ordinary convolutional layers which makes this block essentially an ordinary squeeze and excitation block. This means that the baseline CNNs even have slightly more parameters than the ones with scattering transformation. We clarified this by adjusting the following paragraph in section 2.5 of our manuscript:
> “In the corresponding baseline networks, scattering transformations were replaced with equivalent ordinary learnable convolutional layers. This means that the baseline CNNs have up to 8 k more parameters.”
>
>
> 3) We agree that shape based techniques rely on statistics obtained from data. We therefore removed this statement from the introduction.
>
>
> We again thank Reviewer #4 for his suggestions.

---

### Meta-Review · Area_Chair1 · 2021-03-29

**Recommendation:** Accept (Poster)

**Metareview:**

This paper proposes a new methodological development (combination of scattering transform and squeeze-and-excitation networks) to target small datasets for image segmentation.

On the whole, reviewers find the idea very interesting and the paper clearly written and easy to follow, but wish there were more experiments to back up the claims. The authors produced a thoughful rebuttal and took the reviewer's remarks into account.

I recommend acceptance of this paper.

**Paper Type:**

methodological development

---

### Decision · Program_Chairs · 2021-03-31

Accept